# Immune Reconstitution Inflammatory Syndrome in People Living with HIV Who Presented with Interstitial Pneumonitis: an Emerging Challenge in the Era of Rapid Initiation of Antiretroviral Therapy

Kai-Hsiang Chen,[a] Wang-Da Liu,[a,b] Hsin-Yun Sun,[a] Kuan-Yin Lin,[a,c] Szu-Min Hsieh,[a] Wang-Huei Sheng,[a,d,e] Yu-Chung Chuang,[a] Yu-Shan Huang,[a] Aristine Cheng,[a] Chien-Ching Hung[a,f,g]

aDepartment of Internal Medicine, National Taiwan University Hospital and National Taiwan University College of Medicine, Taipei, Taiwan

bDepartment of Medicine, National Taiwan University Cancer Center, Taipei, Taiwan

cCenter of Infection Control, National Taiwan University Hospital, Taipei, Taiwan

dDepartment of Medical Education, National Taiwan University Hospital and National Taiwan University College of Medicine, Taipei, Taiwan

eSchool of Medicine, National Taiwan University College of Medicine, Taipei, Taiwan

fDepartment of Internal Medicine, National Taiwan University Hospital Yunlin Branch, Yunlin, Taiwan

gDepartment of Tropical Medicine and Parasitology, National Taiwan University College of Medicine, Taipei, Taiwan

**ABSTRACT** Studies on immune reconstitution inflammatory syndrome (IRIS) in people living with HIV (PLWH) and presenting with interstitial pneumonitis (IP) are limited in the era of rapid antiretroviral therapy (ART) initiation, particularly with integrase strand-transfer inhibitor (INSTI)-containing regimens. Adult PLWH presenting with IP in whom ART was initiated within 30 days of IP diagnosis between 2015 and 2021 were retrospectively identified. The primary outcome was the occurrence of IRIS within 30 days after admission. Of 88 eligible PLWH with IP (median age, 36 years; CD4 count, 39 cells/mm$^3$), *Pneumocystis jirovecii* and cytomegalovirus (CMV) DNA were detected via polymerase-chain-reaction assay in 69.3% and 91.7% of respiratory specimens, respectively. 22 PLWH (25.0%) had manifestations that met French's IRIS criteria for paradoxical IRIS. There were no statistically significant differences in terms of the all-cause mortality (0.0% versus 6.1%, $P = 0.24$), the occurrence of respiratory failure (22.7% versus 19.7%, $P = 0.76$), and pneumothorax (9.1% versus 7.6%, $P = 0.82$) between PLWH with and those without paradoxical IRIS. In a multivariable analysis, the factors associated with IRIS were the decline of the 1 month plasma HIV RNA load (PVL) with ART (adjusted hazard ratio [aHR] per 1 log decrease, 3.45; 95% CI, 1.52 to 7.81), a baseline CD4-to-CD8 ratio of <0.1 (aHR, 3.47; 95% CI, 1.16 to 10.44), and the rapid initiation of ART (aHR, 7.95; 95% CI, 1.04 to 60.90). In conclusion, we found a high rate of paradoxical IRIS among PLWH with IP in the era of rapid ART initiation with INSTI-containing ART and this was associated with immune depletion at baseline, a rapid decline of PVL, and an interval of <7 days between the diagnosis of IP and the initiation of ART.

**IMPORTANCE** Our study of PLWH who presented with IP mainly due to *Pneumocystis jirovecii* demonstrates that a high rate of paradoxical IRIS and a rapid decline of PVL with the initiation of ART, a CD4-to-CD8 ratio of <0.1 at baseline, and a short interval (<7 days) between the diagnosis of IP and the initiation of ART were associated with paradoxical IP-IRIS in PLWH. Paradoxical IP-IRIS was not associated with mortality or respiratory failure with heightened awareness among the HIV-treating physicians, rigorous investigations to exclude the possibilities of concomitant infections, or the malignancies and adverse effects of medications, including the cautious use of corticosteroids.

Address correspondence to Chien-Ching Hung, hcc0401@ntu.edu.tw.

The authors declare no conflict of interest.

**KEYWORDS** AIDS, opportunistic infection, opportunistic illness, *Pneumocystis jirovecii*, cytomegalovirus, antiretroviral therapy, integrase strand-transfer inhibitor

Interstitial pneumonitis (IP) is often diagnosed in people living with HIV (PLWH) who have poor adherence to antiretroviral therapy (ART), which results in immunologic failure, or who are late presenters. *Pneumocystis jirovecii* is a leading cause of IP in these PLWH (1), and other etiologies of IP may include cytomegalovirus (CMV), *Haemophilus influenzae*, and respiratory viruses, such as Severe acute respiratory syndrome-Coronavirus 2 (SARS-CoV-2). PLWH presenting with pneumocystis pneumonia (PCP) are treated with trimethoprim-sulfamethoxazole (TMP/SMX), and combination ART is initiated early to improve the outcomes (2, 3). However, some PLWH may initially respond to ART and antimicrobial therapy with improvement, but they may subsequently experience clinical deterioration with fever, hypoxia, dyspnea, or a progression of pulmonary opacification on radiography. In such cases, a diagnosis of paradoxical immune reconstitution inflammatory syndrome (IRIS) is often made after the exclusion of concurrent infections due to other pathogens, inappropriate treatment as the cause of the clinical deterioration, and the adverse effects of medications. Most of the patients recover with the administration of corticosteroids (4).

The reported incidence of IRIS ranged from 5.0% to 40.0% in PLWH responding to ART (5, 6). IRIS is considered to be the restoration of a pathogen-specific immune response with the initiation of ART because of qualitative changes in either lymphocyte function or lymphocyte phenotypic expression (6). The wide range of incidence may have resulted from a lack of consensus on the definitions of IRIS and the different study populations that were included in different settings (7). In a systematic review and meta-analysis, cryptococcal meningitis, progressive multifocal leukoencephalopathy, tuberculosis, and Kaposi's sarcoma were the leading opportunistic illnesses that were reported to be complicated with IRIS (7). Factors that were often demonstrated to be associated with IRIS in previous studies included lower CD4 counts and higher PVL values at baseline (8–10).

In contrast to the opportunistic illnesses that are frequently complicated with IRIS, most studies on IRIS in PLWH presenting with IP due to PCP were case reports (11–14). In a retrospective study by Achenbach et al., 3 of 72 (4.2%) PLWH with PCP developed paradoxical IRIS during the first year on ART (15). In a randomized clinical trial to assess the benefit of early versus deferred ART, it was revealed that PCP-related IRIS developed in 13 of 177 (7.3%) participants; of note, more than 85.0% of the included participants received protease inhibitor (PI)-based ART (2). In another retrospective study in 2021, paradoxical IRIS occurred in 12.4% of PLWH with PCP; however, data on the interval between PCP diagnosis and ART initiation were not available, and most included patients received PI-based ART (67.9%) (16).

Early ART initiation, particularly with integrase strand-transfer inhibitor (INSTI)-based regimens, has currently become the standard of care in international guidelines due to the benefit of improved outcomes in PLWH with opportunistic illnesses and the faster decline of plasma HIV RNA load (PVL) (17–24). However, studies on IP-related IRIS remain scarce in this setting. In this retrospective study, we aimed to examine the incidence of and the factors associated with paradoxical IRIS in PLWH who presented with IP in the setting of rapid ART initiation, particularly with INSTI-based regimens.

## RESULTS

**Characteristics of the included PLWH.** During the 7-year study period, 91 PLWH were diagnosed with IP, with three being excluded due to the lack of baseline CD4 count and PVL value ($n = 1$), lymphoma on chemotherapy ($n = 1$), and no initiation of ART within 30 days ($n = 1$) (Fig. 1). Table 1 shows the baseline clinical characteristics of the included PLWH. Their median baseline CD4 count and PVL values were 39 cells/mm$^3$ (interquartile range [IQR], 8 to 167) and 5.3 log$_{10}$ copies/mL (IQR, 5.1 to 5.8), respectively. The median interval from the diagnosis of IP to the initiation of ART was 2.5 days (IQR, 1 to 7 days).

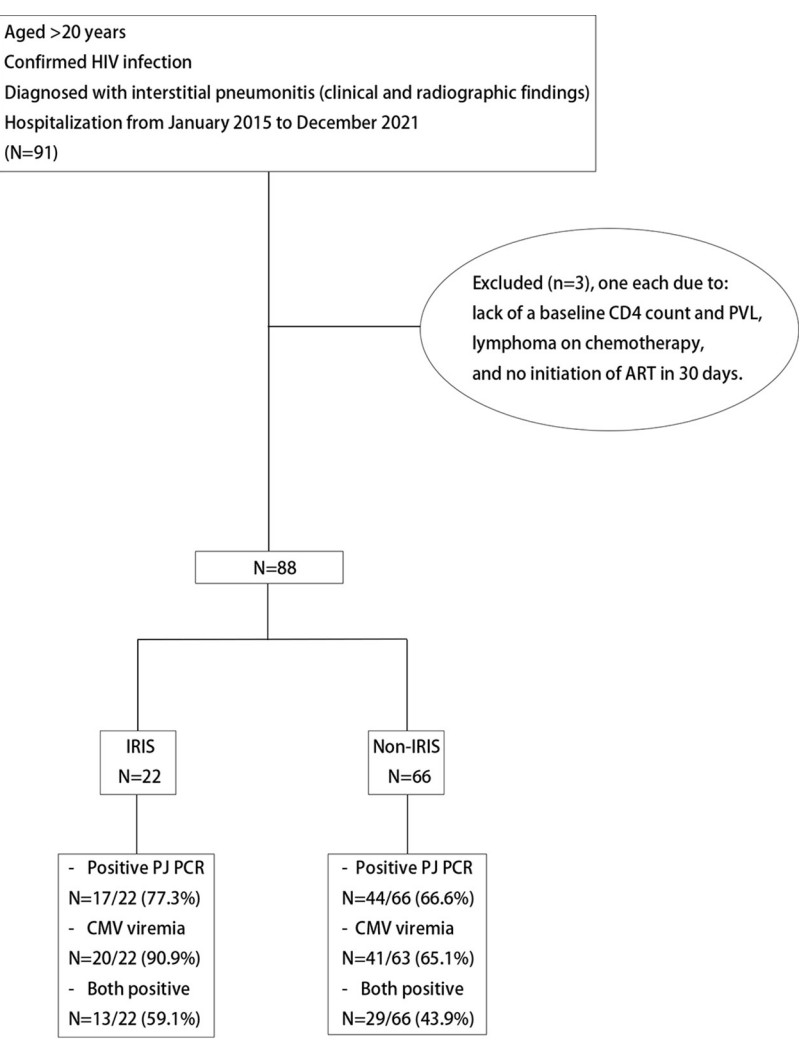

**FIG 1** Study population and investigation of the etiology of interstitial pneumonitis. Abbreviations: CMV, cytomegalovirus; PJ, *Pneumocystis jirovecii*; PVL, plasma HIV RNA load.

**IRIS and treatment of IP and HIV infection.** A diagnosis of paradoxical IRIS was made in 22 (25.0%) of 88 PLWH. Compared with PLWH who did not develop IRIS (non-IRIS group), those who developed IRIS (IRIS group) were more likely to have higher PVL (5.6 versus 5.2 $\log_{10}$ copies/mL, $P = 0.02$) and CD4-to-CD8 ratios of <0.1 (72.7% versus 42.4%, $P = 0.01$) at baseline. All included PLWH received ART after the diagnosis of IP (76.1% within 7 days of IP diagnosis) and two-thirds (67.0%) initiated INSTI-containing regimens. PLWH in the IRIS group tended to have a shorter interval between IP diagnosis and ART initiation than did those without IRIS (1 [IQR, 1 to 3] versus 3 [IQR, 1 to 8] days, $P = 0.01$). There was a numerically higher proportion of PLWH in the IRIS group that started INSTI-based regimens than was observed in the non-IRIS group (81.8% and 65.2%, $P = 0.14$). The interval change of PVL at 1 month of ART was significantly greater for the IRIS group than for the non-IRIS group (5.6 versus 5.2 $\log_{10}$ copies/mL, $P = 0.02$), but the interval increases of the CD4 count were not significantly different between the two groups (91 versus 134 cells/mm³, $P = 0.19$). There were no statistically significant differences between the two groups in terms of age, sex, body-mass index (BMI), CD4 count, CD4-to-CD8 ratio at baseline, or severity of IP at presentation.

*P. jirovecii* was detected via polymerase-chain-reaction (PCR) assay in 61 (69.3%) of the respiratory specimens that were collected from 88 PLWH, whereas CMV DNA was detected in 61 (71.8%) of the serum specimens that were collected from 85 PLWH. A significantly higher proportion of PLWH in the IRIS group had CMV viremia than did

**TABLE 1** Demographic and clinical characteristics of the included people living with HIV who presented with interstitial pneumonitis[a]

| Characteristic | Total | IRIS | Non-IRIS | P value |
|---|---|---|---|---|
| | (N = 88) | (N = 22) | (N = 66) | |
| Age at HIV diagnosis, median (IQR), yrs | 36 (22 to 53) | 33 (25 to 46) | 37 (22 to 50) | 0.48 |
| Male, n (%) | 86 (97.7) | 22 (100.0) | 64 (97.0) | 0.41 |
| Obesity (BMI >27 kg/m$^2$), n (%) | 8 (9.1) | 2 (9.1) | 6 (9.1) | 1.00 |
| MSM/Bisexual, n (%) | 72 (81.8) | 18 (81.8) | 54 (81.8) | 1.00 |
| Laboratory data at IP diagnosis | | | | |
| CD4 count, median (IQR), cells/mm$^3$ | 39 (8 to 167) | 42.5 (13 to 64) | 37 (10 to 156) | 0.68 |
| CD8 count, median (IQR), cells/mm$^3$ | 519 (297 to 765) | 691 (359 to 1083) | 427.5 (285 to 697) | 0.02 |
| CD4-to-CD8 ratio, median (IQR) | 0.1 (0.04 to 0.14) | 0.07 (0.04 to 0.10) | 0.1 (0.05 to 0.16) | 0.06 |
| CD4-to-CD8 ratio <0.1, n (%) | 44 (50.0) | 16 (72.7) | 28 (42.4) | 0.01 |
| PVL, median (IQR), log$_{10}$ copies/mL | 5.3 (5.1 to 5.8) | 5.6 (5.2 to 6.2) | 5.2 (5.0 to 5.7) | 0.01 |
| Absolute neutrophil count, median (IQR), cells/mm$^3$ | 5729 (3488 to 7455) | 5936 (4276 to 7296) | 5523 (3400 to 7455) | 0.77 |
| C-reactive protein (IQR), mg/dL | 3.8 (1.5 to 8.0) | 2.0 (1.2 to 6.0) | 5.1 (1.6 to 11.1) | 0.05 |
| Severity of PCP at presentation | | | | |
| Serum LDH, median (IQR), U/L | 381 (261 to 495) | 416 (307 to 478) | 353 (252 to 507) | 0.42 |
| P/F ratio at presentation, median (IQR), mm Hg | 268.3 (199.0 to 416.7) | 250.0 (215.0 to 381.0) | 283.5 (183.0 to 428.6) | 0.56 |
| ART | | | | |
| Time from ART initiation to IRIS, median (IQR), days | —[b] | 9.5 (6 to 11) | — | — |
| Time from IP diagnosis to ART initiation (IQR), days | 2.5 (1 to 7) | 1 (1 to 3) | 3 (1 to 8) | 0.01 |
| ART initiation <7 days of IP diagnosis, n (%) | 67 (76.1) | 21 (95.5) | 46 (69.7) | 0.01 |
| INSTI-based regimen, n (%) | 59 (67.0) | 18 (81.8) | 43 (65.2) | 0.14 |
| Treatment response to ART | | | | |
| Interval change of PVL at one month of ART initiation, median (IQR), log$_{10}$ copies/mL | 5.3 (5.1 to 5.8) | 5.6 (5.2 to 6.2) | 5.2 (5.0 to 5.7) | 0.02 |
| Interval change of CD4 count at one month of ART initiation, median (IQR), cells/mm$^3$ | 120 (46 to 195) | 91 (26 to 165) | 134 (51 to 199) | 0.19 |
| IP Treatment | | | | |
| TMP/SMX, n (%) | 87 (98.9) | 21 (95.5) | 66 (100.0) | 0.08 |
| Anidulafungin, n (%) | 1 (1.1) | 1 (4.5) | 0 (0.0) | 0.19 |
| Ganciclovir, n (%) | 38 (43.2) | 16 (72.7) | 22 (33.3) | 0.001 |
| Time from IP to ganciclovir treatment, median (IQR), days | 6 (3 to 8) | 6 (4 to 7) | 5 (3 to 8.5) | 0.98 |
| Use of corticosteroid for IP before IRIS, n (%) | 45 (51.1) | 12 (54.5) | 33 (50.0) | 0.71 |
| Prednisone-equivalent dose in the first 5 days, mg/kg/day | 0.9 (0.7 to 1.2) | 0.8 (0.7 to 0.9) | 1.0 (0.7 to 1.2) | 0.22 |
| Microbiologic data | | | | |
| CMV viremia, n (%) | 61/85 (71.8) | 20/22 (90.9) | 41/63 (65.1) | 0.02 |
| CMV detected from respiratory specimens, n (%) | 44/48 (91.7) | 17/18 (94.4) | 27/30 (90.0) | 0.59 |
| | (N = 85) | (N = 22) | (N = 63) | |
| Serum CMV DNA load, median (IQR), log$_{10}$ IU/mL | 2.2 (0.0 to 2.8) | 2.5 (2.2 to 3.1) | 2.2 (0.0 to 2.8) | 0.13 |
| | (N = 48) | (N = 18) | (N = 30) | |
| CMV DNA load in respiratory specimens, median (IQR), log$_{10}$ IU/mL | 4.4 (3.3 to 5.0) | 4.3 (3.7 to 5.0) | 4.4 (3.1 to 5.1) | 0.79 |
| Positive *Pneumocystis jirovecii* DNA PCR in respiratory specimens, n (%) | 88 (69.3) | 17 (77.3) | 44 (66.6) | 0.35 |
| Outcomes | | | | |
| Mortality, n (%) | 4 (4.5) | 0 (0.0) | 4 (6.1) | 0.24 |
| Median time after IP diagnosis (IQR), days | — | — | 22.5 (16 to 27) | — |
| Respiratory failure, n (%) | 18 (20.5) | 5 (22.7) | 13 (19.7) | 0.76 |
| Median time after IP diagnosis (IQR), days | 6 (0 to 9) | 7 (6 to 8) | 0 (0 to 10) | 0.76 |
| Occurrence after IRIS, n (%) | — | 0/5 (0.0) | — | — |
| Pneumothorax, n (%) | 7 (8.0) | 2 (9.1) | 5 (7.6) | 0.82 |
| Median time after IP diagnosis (IQR), days | 9 (4 to 19) | 9.5 (9 to 10) | 9 (4 to 19) | 0.85 |
| Occurrence after IRIS | — | 0/2 (0.0) | — | — |

[a]ART, antiretroviral therapy; BMI, body-mass index; CMV, cytomegalovirus; INSTI, integrase strand-transfer inhibitor; IQR, interquartile range; IP, interstitial pneumonitis; IRIS, immune reconstitution inflammatory syndrome; LDH, lactate dehydrogenase; MSM, men who have sex with men; PVL, plasma HIV RNA load; PCR, polymerase-chain-reaction; P/F, PaO$_2$/FiO$_2$; TMP/SMX, trimethoprim-sulfamethoxazole.
[b]Dashes indicates that there were no applicable data.

PLWH in the non-IRIS group (90.9% versus 65.1%, P = 0.02). However, there was no statistically significant difference between the groups in terms of CMV detected from respiratory specimens (94.4% [17/18] versus 90.0% [27/30], P = 0.59). Of the PLWH with available data on CMV DNA loads, there were no statistically significant differences between the groups in terms of serum (2.5 versus 2.2 log$_{10}$ IU/mL, P = 0.13) or respiratory specimens (4.3 versus 4.4 log$_{10}$ IU/mL, P = 0.79). However, more PLWH in the IRIS group received ganciclovir than those in the non-IRIS group (72.7% versus 33.3%, P = 0.001).

Corticosteroids in tapering doses were used for the treatment of IP in 51.1% of the included PLWH at an equivalent dose of 0.9 mg/kg prednisolone daily in the first 5 days, and this was not found to be significantly different between the two groups (corticosteroid use, IRIS versus non-IRIS, 54.5% versus 50.0%, $P = 0.71$; daily dose, 0.8 mg/kg versus 1.0 mg/kg in the first 5 days, $P = 0.22$).

**Independent factors associated with IRIS and respiratory failure.** In a multivariate Cox regression analysis, the independent factors associated with IRIS events were the interval change of PVL at 1 month of ART (aHR, per 1 log decrease, 3.45; 95% CI, 1.52 to 7.81), a CD4-to-CD8 ratio of <0.1 at baseline (aHR, 3.47; 95% CI, 1.16 to 10.44), and ART initiation within 7 days of the IP diagnosis (aHR, 7.95; 95% CI, 1.04 to 60.90) (Table 2). A sensitivity analysis was performed to evaluate the association between the interval between the diagnosis of IP and the initiation of ART and the development of IRIS. We further found that ART initiation within 4 to 7 days of IP diagnosis was associated with the development of IRIS (Table S1). The use of an INSTI-based regimen (aHR, 2.20; 95% CI, 0.75 to 6.51), corticosteroid use for oxygen desaturation during IP treatment (aHR, 1.13; 95% CI, 0.49 to 2.62), and the total dose of corticosteroids administered during the first 5 days of use (aHR, 1.12; 95% CI, 0.48 to 2.59) were not statistically significantly associated with the development of IRIS. The impact of ART on the occurrence of IRIS is shown in Fig. 2. The INSTI-based regimens did not display an increased risk of IRIS, compared with other regimens (Log-rank test, $P = 0.14$). The presence of CMV viremia showed correlation with IRIS events in a univariate analysis (aHR, 4.56; 95% CI, 1.07 to 19.53), but this was not statistically significant in a multivariate analysis (aHR, 2.53; 95% CI, 0.57 to 11.19) (Table 2).

At the end of the follow-up, 4 PLWH (4.5%) died, 18 (20.5%) developed respiratory failure that required intubation and mechanical ventilation support, and 7 (8.0%) developed pneumothorax (Table 1). The median intervals from IP diagnosis to death, respiratory failure, and pneumothorax were 23.5, 9, and 9 days, respectively. The incidences of complications and the intervals to these three outcomes showed no significant differences between PLWH with and those without IRIS. None of the four fatal cases were considered to be related to

**TABLE 2** Factors associated with paradoxical IRIS in people living with HIV who presented with interstitial pneumonitis[ab]

| Variable | Univariate | | Multivariable | |
|---|---|---|---|---|
| | HR (95% CI) | *P* value | Adjusted HR (95% CI) | *P* value |
| Age at admission, per 1 yr increase | 0.98 (0.93 to 1.03) | 0.55 | —[c] | — |
| Obesity (BMI > 27 kg/m$^2$) | 1.01 (0.24 to 4.34) | 0.99 | — | — |
| Recent CD4 count, per 10 cell/mm$^3$ increase | 1.01 (0.92 to 1.10) | 0.88 | — | — |
| Recent CD8 count, per 10 cell/mm$^3$ increase | 1.01 (1.00 to 1.02) | 0.01 | — | — |
| CD4-to-CD8 ratio < 0.1 | 2.96 (1.16 to 7.58) | 0.02 | 3.47 (1.16 to 10.44) | 0.03 |
| Recent PVL, per 1 log$_{10}$ copies/mL increase | 2.33 (1.21 to 4.50) | 0.01 | — | — |
| Absolute neutrophil count, per 10 cell/mm$^3$ increase | 0.99 (0.99 to 1.00) | 0.78 | — | — |
| C-reactive protein, per 1 mg/dL increase | 0.89 (0.79 to 1.01) | 0.07 | — | — |
| Serum LDH, per 1 U/L increase | 0.99 (0.99 to 1.00) | 0.88 | — | — |
| P/F ratio at presentation, per 1 mm Hg increase | 0.99 (0.99 to 1.00) | 0.70 | — | — |
| Interval change of PVL in a month, per 1 log$_{10}$ copies/mL decrease | 2.34 (1.19 to 4.59) | 0.01 | 3.45 (1.52 to 7.81) | 0.003 |
| Interval between the diagnosis of IP and ART initiation, per 1 day increase | 0.81 (0.69 to 0.95) | 0.01 | — | — |
| ART initiation <7 days of IP diagnosis | 7.84 (1.05 to 58.32) | 0.04 | 7.95 (1.04 to 60.90) | 0.05 |
| Use of corticosteroid for PCP before IRIS | 1.13 (0.49 to 2.62) | 0.77 | — | — |
| Prednisone-equivalent dose in the first 5 days, per 1 mg/kg/day increase | 0.99 (0.99 to 1.00) | 0.74 | — | — |
| INSTI-based regimen | 2.20 (0.75 to 6.51) | 0.15 | — | — |
| CMV viremia | 4.56 (1.07 to 19.53) | 0.04 | 1.86 (0.39 to 8.85) | 0.44 |
| CMV detected from respiratory specimens | 1.60 (0.21 to 12.01) | 0.65 | — | — |
| Serum CMV DNA load, per 1 log$_{10}$ IU/mL increase | 1.39 (0.98 to 1.95) | 0.74 | — | — |
| CMV DNA load in respiratory specimens, per 1 log$_{10}$ IU/mL increase | 1.05 (0.78 to 1.42) | 0.06 | — | — |
| Positive *Pneumocystis jirovecii* DNA PCR in respiratory specimens | 2.43 (0.82 to 7.18) | 0.10 | — | — |

[a]95% CI, 95% confidence interval; ART, antiretroviral therapy; CMV, cytomegalovirus; HR, hazard ratio; INSTI, integrase strand-transfer inhibitor; IP, interstitial pneumonitis; LDH, lactate dehydrogenase; PVL, plasma HIV RNA load; PCP, pneumocystis pneumonia.

[b]The covariates of PVL, recent CD8 count, and interval between the diagnosis of IP and ART initiation were not included in the multivariable model because there were the potential confounders.

[c]Dashes indicates that the variables in the univariable analysis were not included in multivariable analysis.

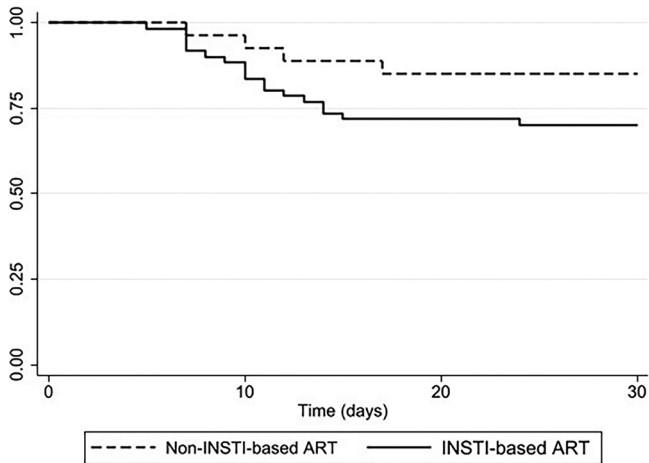

**FIG 2** Kaplan-Meier curve for IRIS among the included people living with HIV, initiating or not initiating INSTI-based regimens (Log-rank test, *P* = 0.14). Abbreviations: INSTI, integrase strand-transfer inhibitor; IRIS, immune reconstitution inflammatory syndrome.

IRIS by the treating physicians, and no case developed respiratory failure after IRIS. In the IRIS group, the duration of hospital stay was significantly longer than that observed in the non-IRIS group (21.5 versus 15 days, *P* = 0.001). The factors that were independently associated with respiratory failure that led to intubation and mechanical ventilation support were a BMI of >27 kg/m² (aHR, 7.83; 95% CI, 1.89 to 32.49) and the absolute neutrophil count (aHR, per 10-cell/mm³ increase, 1.003; 95% CI, 1.001 to 1.005) (Table S2).

In the subgroup analyses of PLWH with *P. jirovecii* identified in respiratory specimens, the findings were similar to those of our primary analyses. A diagnosis of IRIS was made in 17 (27.9%) of 61 PLWH with PCP. In a multivariate Cox regression analysis, the independent factors associated with IRIS events were the interval change of PVL at 1 month of ART (aHR, per 1 log decrease, 2.63; 95% CI, 1.05 to 6.59) and a CD4-to-CD8 ratio of <0.1 at baseline (aHR, 3.39; 95% CI, 1.04 to 11.05) (Table 3). The incidences of complications, including the all-cause mortality (0.0% versus 9.1%, *P* = 0.20), the occurrence of respiratory failure, (29.4% versus 29.5%, *P* = 0.99), and pneumothorax (11.8% versus 11.4%, *P* = 0.97), also showed no significant difference between PLWH with and those without IRIS.

**TABLE 3** Factors associated with IRIS in people living with HIV who presented with interstitial pneumonitis and had *Pneumocystis jirovecii* identified from respiratory specimens[a]

| Variable | Univariate | | Multivariate | |
|---|---|---|---|---|
| | HR (95% CI) | *P* value | Adjusted HR (95% CI) | *P* value |
| Age at admission, per 1 yr increase | 0.99 (0.94 to 1.05) | 0.83 | —[b] | — |
| Obesity (BMI > 27 kg/m²) | 1.08 (0.25 to 4.71) | 0.92 | — | — |
| Recent CD4 count, per 10 cell/mm³ increase | 1.02 (0.94 to 1.12) | 0.60 | — | — |
| Recent CD8 count, per 10 cell/mm³ increase | 1.01 (1.00 to 1.02) | 0.01 | — | — |
| CD4-to-CD8 ratio < 0.1 | 2.51 (0.88 to 7.12) | 0.08 | 3.39 (1.04 to 11.05) | 0.04 |
| Absolute neutrophil count, per 10 cell/mm³ increase | 0.99 (0.99 to 1.00) | 0.68 | — | — |
| C-reactive protein, per 1 mg/dL increase | 0.88 (0.77 to 1.01) | 0.06 | — | — |
| Serum LDH, per 1 U/L increase | 0.99 (0.99 to 1.00) | 0.92 | — | — |
| P/F ratio at presentation, per 1 mm Hg increase | 0.99 (0.99 to 1.00) | 0.91 | | |
| Recent PVL, per 1 log₁₀ copies/mL increase | 2.05 (0.97 to 4.33) | 0.06 | — | — |
| Interval change of PVL in a month, per 1 log₁₀ copies/mL decrease | 1.93 (0.88 to 4.21) | 0.10 | 2.63 (1.05 to 6.59) | 0.04 |
| Interval between the diagnosis of IP and ART initiation, per 1 day increase | 0.85 (0.72 to 1.01) | 0.06 | 0.87 (0.75 to 1.02) | 0.08 |
| ART initiation <7 days of IP diagnosis | 5.23 (0.69 to 39.46) | 0.11 | — | — |
| Prednisone-equivalent dose in the first 5 days, per 1 mg/kg/day increase | 0.33 (0.06 to 1.72) | 0.19 | — | — |
| INSTI-based regimen | 1.28 (0.27 to 6.02) | 0.76 | — | — |
| IRIS | 1.59 (0.52 to 4.86) | 0.42 | — | — |
| CMV viremia | 3.49 (0.80 to 15.28) | 0.10 | — | — |

[a]Abbreviations: 95% CI, 95% confidence interval; BMI, body-mass index; CMV, cytomegalovirus; HR, hazard ratio; IRIS, immune reconstitution inflammatory syndrome; LDH, lactate dehydrogenase; PVL, plasma HIV RNA load; P/F, PaO₂/FiO₂.
[b]Dashes indicates that the variables in the univariable analysis were not included in multivariable analysis.

## DISCUSSION

This study demonstrates that the overall rate of IP-related paradoxical IRIS was estimated to be 25% in the setting of rapid ART initiation. We found that a greater PVL decline at 1 month of ART initiation, a baseline CD4-to-CD8 ratio of <0.1, and a shorter interval (<7 days) between the diagnosis of IP and the initiation of ART were statistically significantly associated with the development of paradoxical IRIS. Moreover, we found that obesity and a higher absolute neutrophil count at baseline were associated with respiratory failure that required mechanical ventilation.

Studies investigating IP-related paradoxical IRIS among PLWH are limited in the era of early ART initiation with INSTI-based regimens (11–13). Several factors have been found to be associated with the development of IRIS in previous studies before the widespread use of INTSI-containing regimens, which included a nadir CD4 count of <100 cells/mm$^3$, a baseline CD4 cell percentage of <10%, a younger age, a PVL decrease of more than 2.5 $\log_{10}$ copies/mL at the time of IRIS, and the use of a boosted PI-based regimen (compared with a non-nucleoside reverse transcriptase inhibitor-based regimen) (8–10, 25). Consistent with the findings of previous studies (8–10), we found that PLWH with IP who had a greater 1 month PVL decline with ART were at an increased risk for paradoxical IRIS when ART was started. However, our study revealed a higher rate (25.0%) of paradoxical IRIS during IP treatment than did previous studies (5 to 12.4%) (2, 16, 26). The reasons for the discrepancies in terms of the incidence of paradoxical IP-related IRIS that were observed between our study and other studies could be attributed to the IRIS definition that was used, the etiologies of IP, the timing of ART initiation, and the ART regimens initiated (2, 16, 26).

In our study, the interval between the diagnosis of IP and ART initiation was 7 days or less in 76.1% of the included PLWH presenting with IP. We found that a shorter interval (<7 days) between the diagnosis of IP and the initiation of ART was associated with an increased risk of developing paradoxical IRIS, which might be contributory to the higher rate of PLWH developing IP-related paradoxical IRIS that was observed in our study. However, in a randomized study comparing early versus deferred ART initiation, the early initiation of ART (immediate after enrollment versus >4 weeks after enrollment) did not increase the incidence of IRIS (8). The discrepant results may be due to ART being initiated early with a median interval of 2.5 days after the treatment of opportunistic illness was begun in our study, compared to a median interval of 12 days in the previous study. Early ART initiation (<14 days) has been proven to be associated with reduced mortality among PLWH who present with opportunistic illnesses, except for the presence of cryptococcal or tuberculosis meningitis (2, 27, 28). While early ART initiation improves the outcomes of PLWH (2), increased occurrences of IRIS could pose challenges to clinical management, particularly in regions where resources with which to differentiate clinical manifestations from those of IRIS are limited due to concurrent opportunistic illnesses, antimicrobial resistance, and adverse drug reactions.

In our study, 67.0% of the included PLWH initiated INSTI-based regimens that have been demonstrated to have faster PVL declines and greater CD4 increases than other ART classes (18, 29, 30). This has raised concerns about an increased risk for IRIS in PLWH who initiate INSTI-containing ART at advanced stages of HIV infections. However, clinical studies yielded conflicting results due to the different study populations, designs, and microbiologic investigations that were performed to exclude other causes for clinical deterioration. For example, Kityo et al. showed no difference between INSTI- and non-INSTI-based regimens in terms of IRIS in a randomized clinical trial, whereas the observational study by Wijting et al. found a higher risk of developing IRIS for PLWH who received an INSTI-based regimen (31). In our study, which included PLWH who started ART early with a high proportion of the PLWH being on INSTI-containing regimens, we failed to demonstrate a statistically significant association between paradoxical IRIS and the use of INSTI-containing regimens, likely due to the small sample size of our study.

In our study, high proportions of the included PLWH had CMV DNA that was detectable via PCR assay in serum (72%) and respiratory (92%) specimens, and 43.2% of the included

PLWH who presented with IP also received ganciclovir. While CMV is a common pathogen isolated from respiratory specimens in PLWH, and while the presence of CMV may suggest an advanced HIV infection and a dampened immune status, the pathogenic role of CMV remains debated, particularly in PLWH with clinical deterioration of IP, despite treatment with TMP/SMX. Several studies have shown that the mortality and severity of PCP were not affected by anti-CMV therapy in PLWH with shedding of CMV in the respiratory tract (32–35). While CMV was highly prevalent among the included PLWH in our study, the role of CMV in paradoxical IRIS or pneumonitis when clinical deterioration occurs warrants further investigation.

In general, IRIS is a self-limited complication after ART initiation. In some instances, adjunctive corticosteroids may be needed for PLWH who experience severe symptoms (4). Overall, the mortality rate in PLWH with IRIS is not higher than that observed among those without IRIS, except for IRIS affecting the central nervous system (5, 36, 37), such as PLWH with cryptococcal meningitis-associated IRIS, in whom the mortality rate could be as high as 20%. In our study, the four fatal cases occurred in the non-IRIS group. We did not find that respiratory failure requiring intubation and mechanical ventilation or pneumothorax were related to paradoxical IRIS. However, the length of hospital stay was still significantly longer in our PLWH with IRIS due to the investigations and management required for IRIS, which was concordant with the results of Kann's study that focused on PCP-related IRIS (16).

Our study has several limitations. First, this retrospective study included only those with *P. jirovecii* and CMV identified in the respiratory specimens, and the findings observed might not be generalizable to PLWH with IP due to other etiologies. Second, most of the PLWH who were included in our study used sputum or gargling specimens to detect *P. jirovecii* with the use of PCR assays. In a meta-analysis, minimally invasive PCP diagnostic tests that used induced sputum and oral wash showed good specificities in PLWH (98% and 95%, respectively) (38); however, the oral wash specimens had a lower sensitivity (74%). Therefore, the diagnosis of PCP in our study might have been underestimated. In a subgroup analysis that was limited to PLWH with *P. jirovecii* identified from respiratory specimens, we found similar results, regarding the primary and secondary outcomes as well as the factors that were associated with paradoxical IRIS, to those of the primary analyses that were presented. Third, histopathologic confirmation of CMV pneumonitis was not made both before and after clinical deterioration to better define the role of CMV at the onset of IRIS, despite the findings that CMV was highly prevalent in the respiratory and blood specimens in our study. Fourth, the sample size was relatively small, which might preclude us from evaluating whether INSTI-based regimens increase the incidence of IRIS in the setting of rapid ART initiation. Fifth, because we only evaluated the occurrence of paradoxical IRIS, the conclusions of our study could not be generalized to PLWH with IP due to unmasking IRIS.

In conclusion, we found a high rate of IP-related paradoxical IRIS in the era of rapid ART, particularly with integrase inhibitor-based regimens. A greater decline of PVL at 1 month of ART initiation, a low CD4-to-CD8 ratio (<0.1) at baseline, and a shorter interval (<7 days) between the diagnosis of IP and the initiation of ART were associated with the development of paradoxical IRIS in PLWH who presented with IP. PLWH presenting with IP who were obese and had an elevated absolute neutrophil count were more likely to develop respiratory failure and require mechanical ventilator support.

## MATERIALS AND METHODS

**Study design and setting.** This retrospective cohort study was conducted at the National Taiwan University Hospital to include PLWH aged 20 years or older who presented with IP from January of 2015 to December of 2021. Only those with positive PCR results that identified *P. jirovecii* and CMV from either serum or respiratory specimens were included. PLWH with solid tumors who were undergoing immunosuppressive therapy, without baseline and follow-up CD4 counts or PVL data, or without ART initiation within 30 days of the IP diagnosis were excluded. Medical records were retrospectively reviewed to obtain demographic and clinical information, which included the age, sex, BMI, route of HIV transmission, ART and other medications, blood examinations, microbiological investigations, severity of IP at presentation ($PaO_2/FiO_2$ ratio and serum lactate dehydrogenase [LDH] level), death, respiratory failure

that required intubation and mechanical ventilation support, and development of pneumothorax of the patients. The study was approved by the Research Ethics Committee of the hospital (202210035RINB).

PLWH with AIDS-related opportunistic illnesses were recommended to initiate ART as early as possible, according to the suggestion of the World Health Organization (39). Since 2016, all PLWH were recommended to initiate ART, regardless of CD4 count, in Taiwan. Coformulated, single-tablet ART regimens were introduced into Taiwan after 2016 as the first-line ART, and they were provided to PLWH free-of-charge. These ART regimens included tenofovir disoproxil fumarate (TDF)/emtricitabine (FTC)/efavirenz and TDF/FTC/rilpivirine, abacavir/lamivudine/dolutegravir in 2016; tenofovir alafenamide (TAF)/FTC/rilpivirine and TAF/FTC/elvitegravir/cobicistat in 2018; and TAF/FTC/bictegravir and lamivudine/dolutegravir in 2019 and 2020, respectively. The national HIV treatment guidelines recommended ART initiation within 7 days of a confirmed HIV diagnosis in 2018 and same-day ART initiation in 2021 (40, 41). Testing for CD4 and PVL are performed at baseline, at 1 month of ART initiation, subsequently every 3 months within the first year of ART initiation, and every 3 to 6 months after the PLWH have achieved viral suppression on stable ART.

**Management of PLWH with IP.** The diagnosis of IP was defined by the presence of (i) at least two of the three following symptoms/signs of cough, fever, and dyspnea, and (ii) radiographic findings of diffuse, bilateral, and interstitial infiltrates on chest radiography or ground-glass opacities on the computed tomography of the chest.

The investigations of the etiology of IP included the collection of sputum or gargling specimens for PCR assays for the detection of *P. jirovecii* (BD MAX system; Beckon Dickinson, Diagnostic Systems, Sparks, MD, USA) as well as serum and/or sputum or gargling specimens for CMV DNA (cobas CMV; Roche Molecular Systems, Inc., Branchburg, NJ, USA) and microbiologic cultures, serum specimens for cryptococcal and aspergillus antigen assays, and sputum specimens for acid-fast smears, mycobacterial cultures, and nucleic-acid testing (GeneXpert MTB/RIF Ultra assay; Cepheid Inc, USA). Bronchoscopy with bronchoalveolar lavage or biopsy specimen was performed if the aforementioned tests failed to provide an etiologic diagnosis. In some instances, video-assisted thoracoscopy was performed to obtain lung tissue for pathology and microbiologic cultures.

All of the included PLWH received a therapeutic dose of TMP/SMX (TMP, 10 to 12 mg/Kg/day, divided into three doses) empirically when the diagnosis of IP was made based on a clinical assessment and radiographic findings. Those with a glucose-6-phosphate dehydrogenase deficiency or a developing allergy to TMP/SMX were treated with anidulafungin as an alternative treatment (42). Adjunctive corticosteroids were prescribed for those with oxygen desaturation at an initial dose of 40 mg of oral prednisone twice daily on days 1 to 5 and 40 mg of prednisone daily on days 6 to 10. This was followed by 20 mg of prednisone daily on days 11 to 21 (3). For those with a positive PCR result for CMV from a serum or respiratory specimen, ganciclovir was prescribed at the physician's discretion for those with oxygen desaturation if CMV pneumonitis was suspected.

**Definitions of paradoxical IRIS.** Diagnoses of paradoxical IRIS events was made according to French's case criteria, which are based on 2 major and 3 minor criteria (28). The first major criterion (i.e., the atypical presentation of opportunistic infections or tumors in patients responding to ART) is essential and must be accompanied by evidence of either a therapeutic response to ART with a decrease in HIV PVL by $>1 \log_{10}$ copies/mL or by at least two minor criteria. The minor criteria include an increase in the CD4 cell count, a spontaneous resolution of symptoms, or a measurable, pathogen-specific immune response (43).

**Management of PLWH with paradoxical IP-IRIS.** Comprehensive investigations, including microbiology (bacterial, mycobacteria, fungus, and CMV) and inflammatory markers (CRP and LDH), were performed for every PLWH with IP who presented with oxygen desaturation, new-onset fevers, or pulmonary infiltrates. Once the diagnosis of paradoxical IRIS was made, according to French's criteria, adjunctive corticosteroids with a methylprednisolone dose of 0.5 to 1 mg/kg/day was administered (11). The dose of corticosteroids was tapered every 3 to 5 days until defervescence, an improvement of oxygen desaturation, and a resolution of pulmonary opacities were achieved.

**Outcome assessment.** Patients were followed-up for the development of paradoxical IRIS from the diagnosis of IP. Each patient was observed until death or 30 days after the diagnosis of IP. The primary outcome was the occurrence of paradoxical IRIS, whereas secondary outcomes included the 30-day all-cause mortality, occurrence of respiratory failure that required intubation and mechanical ventilation support, and development of pneumothorax.

**Statistical analysis.** Categorical variables were compared between PLWH with and those without IRIS events via chi-squared tests. Continuous variables were analyzed using Wilcoxon rank-sum tests. The Kaplan-Meier curve with the log-rank test was used to demonstrate the occurrences of IRIS between PLWH receiving INSTI-based regimens and PLWH receiving other ART regimens. Cox proportional hazards models were used to estimate the unadjusted and adjusted hazard ratios (aHRs) for IRIS. We used stepwise Cox regression with a removal threshold of $P = 0.05$ to select among the covariates to be included in the multivariable model. A sensitivity analysis was performed to evaluate the association of the interval between the diagnosis of IP and the initiation of ART and the development of paradoxical IRIS. All tests were two-tailed, and a $P$ value of $<0.05$ was considered to be indicative of a statistically significant result. All prespecified tests were also performed in the subgroup only, including the PLWH with confirmed PCP who had positive *P. jirovecii* DNA PCR results for their respiratory specimens. All analyses were performed using the STATA/SE software package, Version 17.0 (http://www.stata.com).

**Data availability.** Deidentified participant-level data will be available upon the publication of the study. Requests for data should be sent to Chien-Ching Hung at the e-mail address in the corresponding author footnote. Upon the review of the proposed protocol and the signing of a data sharing agreement, the data will be made available. The protocol and consent form will also be available upon email request.

## SUPPLEMENTAL MATERIAL

Supplemental material is available online only.
**SUPPLEMENTAL FILE 1**, PDF file, 0.6 MB.

## ACKNOWLEDGMENT

The authors have no conflict of interest to declare.

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
