## [Reviewer comments · Microbiology Spectrum]

Microbiology Spectrum

Immune reconstitution inflammatory syndrome in people living with HIV who presented with interstitial pneumonitis: an emerging challenge in the era of rapid initiation of antiretroviral therapy

Kai-Hsiang Chen, Wang-Da Liu, Hsin-Yun Sun, Kuan-Yin Lin, Szu-Min Hsieh, Wang-Huei Sheng, Yu-Chung Chuang, Yu-Shan Huang, Aristine Cheng, and Chien-Ching Hung

Corresponding Author(s): Chien-Ching Hung, National Taiwan University Hospital

Review Timeline:

Submission Date:	December 5, 2022
Editorial Decision:	December 27, 2022
Revision Received:	February 8, 2023
Accepted:	February 16, 2023

Editor: Yuan Pin Hung

Reviewer(s): Disclosure of reviewer identity is with reference to reviewer comments included in decision letter(s). The following individuals involved in review of your submission have agreed to reveal their identity: Rachel Lai (Reviewer #1)

Transaction Report:

DOI: <https://doi.org/10.1128/spectrum.04985-22>

December 27, 2022

Dr. Chien-Ching Hung
National Taiwan University Hospital
Department of Internal Medicine
7, Chung-Shan South Road
Taipei 100
Taiwan

Re: Spectrum04985-22 (Immune reconstitution inflammatory syndrome in people living with HIV who presented with interstitial pneumonitis: an emerging challenge in the era of rapid initiation of antiretroviral therapy)

Dear Dr. Chien-Ching Hung:

Link Not Available

Sincerely,

Yuan Pin Hung

Journals Department
Reviewer comments:

Reviewer #1 (Comments for the Author):

The manuscript by Chen et al. documented the incidence of TB-IRIS interstitial pneumonitis (IP) in HIV-TB patients following rapid initiation of antiretroviral therapy. Although the finding itself is interesting, pitifully the knowledge is not new and the study also lacks any experimental work to further characterise IP, which has not been studied in detail and would have provided novel insight.

As the study is observational and might not have clinical specimens available, the authors could consider performing secondary

analysis such as whether time to IP or severity of IP is associated with CD4, CD8 or ratio of CD4/CD8, or any inflammatory markers (e.g. CRP), or abnormalities of any radiographic / imaging findings, and whether different etiologies of IP led to differential outcomes (e.g. time of IRIS, duration of IRIS, duration of treatment or hospitalization, mortality).

Reviewer #2 (Comments for the Author):

This study retrospectively analyzed clinical importance of IRIS who had already had interstitial pneumonia. As authors described, clinical data obtained in this study are very important in daily practice in the era of rapid initiation of ART.

However, I have a couple of major concerns on this study.

First, in the eligible 88 IP patients, P jirovecii was detected in 69.3% and CMV in 91.7%. The patients having P jirovecii could be diagnosed as PCP. However, Patients with CMV DNA could not be diagnosed as definite CMV pneumonitis. How to treat and prognosis of IRIS are quite different on each pathogen. We have often experienced that some PCP-IRIS were much more severe than other pathogens and sometimes fatal. If authors much focused on PCP and PCP-IRIS, it might be possible to obtain much clearer message on how to manage PCP-IRIS.

Second, authors emphasized impact of INSTI-based regimen on IRIS. It should be important. However, in this study, use of INSTI-based regimen had not significantly affected the impact of IRIS. Furthermore, authors described in Line 71 to 76 that rapid decline of HIV RNA and baseline low CD4/CD8 ratio were the important clinical information. However, this information has already been well known and not new. On the other hand, interval between the diagnosis of IP and ART initiation, per 1-day increase had significantly decrease the risk of IRIS. Authors should more emphasize this result in the era of rapid ART in "importance" (Line 64).

Minor comments

1. Line 99; authors described two types of IRIS. In this study only paradoxical IRIS was analyzed. Authors may clearly state that this study did not analyze the unmasking IRIS, because clinical features of paradoxical IRIS and unmasking one are different.
2. Line 54; Twenty-two had manifestations of IRIS. Line 55 and 56; four died and 18 had required mechanical ventilation in IRIS group and non-IRIS group. The description gave us some confusion. Severe cases should be clearly divided into IRIS group and non-IRIS group.

Staff Comments:

Preparing Revision Guidelines

Please return the manuscript within 60 days; if you cannot complete the modification within this time period, please contact me. If you do not wish to modify the manuscript and prefer to submit it to another journal, please notify me of your decision immediately so that the manuscript may be formally withdrawn from consideration by Microbiology Spectrum.

Editor-in-Chief

Microbiology Spectrum

08 Feb, 2023

Manuscript Number: Spectrum04985-22R1

Article Title: Immune reconstitution inflammatory syndrome in people living with HIV who presented with interstitial pneumonitis: an emerging challenge in the era of rapid initiation of antiretroviral therapy

Dear Editor,

Thank you for giving us the opportunity to revise our manuscript. We appreciate the comments and suggestions of the editor and reviewers to improve the quality of our manuscript. The manuscript is revised according to the comments and suggestions of the editor and the reviewers. The specific style requirements are followed and responses to the comments of the editorial office are incorporated into the revised manuscript. Our point-to-point responses to the comments of the reviewers are listed below. Again, we sincerely appreciate your kind consideration of our manuscript for publication in "Microbiology Spectrum".

Best regards

Chien-Ching Hung, M.D., Ph.D.

Department of Internal Medicine, National Taiwan University Hospital,

7 Chung-Shan South Rd., Taipei City 10002, Taiwan

E-mail address: hcc0401@ntu.edu.tw

Tel: +886-2-23123456 ext. 67552

Major comments:

Reviewer #1:

The manuscript by Chen et al. documented the incidence of TB-IRIS interstitial pneumonitis (IP) in HIV-TB patients following rapid initiation of antiretroviral therapy. Although the finding itself is interesting, pitifully the knowledge is not new and the study also lacks any experimental work to further characterise IP, which has not been studied in detail and would have provided novel insight.

1. As the study is observational and might not have clinical specimens available, the authors could consider performing secondary analysis such as whether time to IP or severity of IP is associated with CD4, CD8 or ratio of CD4/CD8, or any inflammatory markers (e.g. CRP), or abnormalities of any radiographic / imaging findings.

Response: Thank you for the comment. We add the related information in the revised manuscript. The study was retrospectively conducted and the recall bias would be significant as to the information on the time to IP. Instead, we used development of respiratory failure to indicate the severity of IP and attempted to analyze the factors associated with respiratory failure that led to intubation and mechanical ventilation support (Supplementary Table 2). The rate of respiratory failure was similar between the patients with and those without IRIS (22.7% vs 19.7%, $p=0.76$) (Table 1). We found factors associated with respiratory failure were obese and an elevated absolute neutrophil count. The CD4 count, CD4/CD8 ratio and CRP level were not associated with development of respiratory failure (Supplementary Table 2). The revision made to “**Table 1 and Supplementary Table 2**” is as follows: The median CRP level was 3.8 mg/dL (IQR, 1.5-8.0 mg/dL) and CRP level was not statistically significantly associated with the development of respiratory failure in univariable analysis (aHR, 0.96; 95% CI 0.75-1.24).

2. Whether different etiologies of IP led to differential outcomes (e.g. time of IRIS, duration of IRIS, duration of treatment or hospitalization, mortality).

Response: Thank you for the comment. In our study, PLWH who were diagnosed with pneumonia due to known pathogens other than *Pneumocystis jirovecii* or cytomegalovirus (CMV) were not included. Indeed, the role of CMV in IP among people living with HIV remains to be defined; therefore, we do subgroup analyses of PLWH with *P. jirovecii* identified from respiratory specimens by polymerase-chain reaction assay, which yield similar findings as our primary analyses (Table 3). The revision made to “**RESULTS**” is as follows: “In the subgroup analyses of PLWH with *P. jirovecii* identified from respiratory specimens, the findings were similar to those in our primary analyses. A diagnosis of IRIS was made in 17 (27.9%) of 61 PLWH with PCP. In the multivariate Cox regression analysis, independent factors associated with IRIS events were the interval change of PVL at 1 month of ART (aHR, per 1-log decrease, 2.63; 95% CI, 1.05-6.59), and a CD4-to-CD8 ratio <0.1 at baseline (aHR, 3.39; 95% CI, 1.04–11.05) (**Table 3**). The incidences of complications, including all-cause mortality (0.0% vs 9.1%, p=0.20), occurrence of respiratory failure, (29.4% vs 29.5%, p=0.99), and pneumothorax (11.8% vs 11.4%, p=0.97), also showed no significant difference between PLWH with and those without IRIS.” (Page 11-12, RESULTS, Independent factors associated with IRIS and respiratory failure, lines 193-202).

Reviewer #2:

This study retrospectively analyzed clinical importance of IRIS who had already had interstitial pneumonia. As authors described, clinical data obtained in this study are very important in daily practice in the era of rapid initiation of ART. However, I have a couple of major concerns on this study.

1. First, in the eligible 88 IP patients, *P. jirovecii* was detected in 69.3% and CMV in 91.7%. The patients having *P. jirovecii* could be diagnosed as PCP. However,

Patients with CMV DNA could not be diagnosed as definite CMV pneumonitis. How to treat and prognosis of IRIS are quite different on each pathogen. We have often experienced that some PCP-IRIS were much more severe than other pathogens and sometimes fatal. If authors much focused on PCP and PCP-IRIS, it might be possible to obtain much clearer message on how to manage PCP-IRIS.

Response: Thank you for the comment. We acknowledge that the role of cytomegalovirus (CMV) was not clear in these population as described in our manuscript. In order to generalize our study results to PLWH with *Pneumocystis* pneumonia (PCP), we performed subgroup analyses to include PLWH with *Pneumocystis jirovecii* identified from respiratory specimens and we found the similar results of primary and secondary outcomes and the factors associated with IRIS between the overall and subgroup analyses (Table 3). While the diagnosis of PCP-IRIS may remain to be more clearly defined, IRIS itself is a diagnosis by exclusion. Therefore, the diagnosis of PCP-IRIS in this study was made only after rigorous investigations to exclude the presence of concomitant infections or malignancies and adverse effects of antimicrobials or other medicines the individuals were receiving. To provide a much clear message on management of PCP-IRIS, we add “Management of PLWH with paradoxical IP-IRIS” in “Materials and Methods” to describe how to manage these patients at our hospital, including a comprehensive workup for microbiology and inflammation markers, the diagnosis of IRIS via French’s criteria, and the strategy of use of steroids. The revision made to “**MATERIALS AND METHODS**” is as follows: “Comprehensive investigations, including microbiology (bacterial, mycobacteria, fungus, and CMV) and inflammatory markers (CRP and LDH) were performed for every PLWH with IP who presented with oxygen desaturation, new-onset fevers or pulmonary infiltrates. Once the diagnosis of paradoxical IRIS was made according to the French’s criteria, adjunctive corticosteroids with the dose of methylprednisolone 0.5-1 mg/kg/day will be administered. The dose of

corticosteroids will be tapered every 3-5 days until defervescence, improvement of oxygen desaturation and resolution of pulmonary opacities were achieved.” (Page 22, Materials and Methods, Management of PLWH with paradoxical IP-IRIS, lines 389-397).

2. Second, authors emphasized impact of INSTI-based regimen on IRIS. It should be important. However, in this study, use of INSTI-based regimen had not significantly affected the impact of IRIS. Furthermore, authors described in Line 71 to 76 that rapid decline of HIV RNA and baseline low CD4/CD8 ratio were the important clinical information. However, this information has already been well known and not new. On the other hand, interval between the diagnosis of IP and ART initiation, per 1-day increase had significantly decrease the risk of IRIS. Authors should more emphasize this result in the era of rapid ART in "importance" (Line 64).

Response: Thank you for the comment. Your comment is highly appreciated. Indeed, several factors such as use of an INSTI-based regimen (aHR, 2.20; 95% CI 0.75-6.51), corticosteroid use for oxygen desaturation during IP treatment (aHR, 1.13; 95% CI 0.49-2.62), and the total dose of corticosteroids administered in the first 5 days of use (aHR, 1.12; 95% CI 0.48-2.59) were found to be associated with the development of paradoxical IP-IRIS in univariable analysis, but not in multivariable analysis, probably due to the small sample size of the included PLWH with IP. The interval associated with the development of paradoxical IP-IRIS in subgroup analysis is shown in **Supplementary Table 1**. The revision made to “**IMPORTANCE**” is as follows: “Our study of people living with HIV (PLWH) who presented with interstitial pneumonitis (IP) mainly due to *Pneumocystis jirovecii* demonstrate that a high rate of paradoxical immune reconstitution inflammatory syndrome (IRIS), and a rapid decline of HIV RNA with initiation of antiretroviral therapy and a CD4-to-CD8 ratio <0.1 at baseline, and a

shorter interval (<7 days) between the diagnosis of IP and ART initiation were associated with paradoxical IP-IRIS in PLWH. Paradoxical IP-IRIS was not associated with mortality or respiratory failure with heightened awareness among the HIV-treating physicians, rigorous investigations to exclude the possibilities of concomitant infections or malignancies and adverse effects of medications, and cautious use of corticosteroids.” (Page 4, **IMPORTANCE**, lines 67-75).

3. Line 99; authors described two types of IRIS. In this study only paradoxical IRIS was analyzed. Authors may clearly state that this study did not analyze the unmasking IRIS, because clinical features of paradoxical IRIS and unmasking one are different.

Response: Thank you for the comment. We revise the section of Materials and Methods to clearly indicate the outcome assessment of our study. The revision made to “**MATERIALS AND METHODS, outcome assessment**” is as follows: “Patients were followed-up for the development of paradoxical IRIS from the diagnosis of IP.” (Page 22, Materials and Methods, outcome assessment, line 389). In Discussion, we describe it as a limitation in that our study only analyzed the included patients with paradoxical IRIS. The revision made to “**DISCUSSION**” is as follows: “Fifth, because we only evaluated the occurrence of paradoxical IRIS, the conclusion of our study could not be generalized to in PLWH with IP due to unmasking IRIS.” (Page 17, Discussion, paragraph7, lines 299-301).

4. Line 54; Twenty-two had manifestations of IRIS. Line 55 and 56; four died and 18 had required mechanical ventilation in IRIS group and non-IRIS group. The description gave us some confusion. Severe cases should be clearly divided into IRIS group and non-IRIS group.

Response: Thank you for the comment. The revision made to “**ABSTRACT**” is as

follows : “There were no statistically significant differences in terms of all-cause mortality (0.0% vs 6.1%, $p=0.24$), occurrence of respiratory failure (22.7% vs 19.7%, $p=0.76$), and pneumothorax (9.1% vs 7.6%, $p=0.82$) between PLWH with and those without paradoxical IRIS.” (Page 3, Abstract, lines 54-57).

February 16, 2023

Dr. Chien-Ching Hung
National Taiwan University Hospital
Department of Internal Medicine
7, Chung-Shan South Road
Taipei 100
Taiwan

Re: Spectrum04985-22R1 (Immune reconstitution inflammatory syndrome in people living with HIV who presented with interstitial pneumonitis: an emerging challenge in the era of rapid initiation of antiretroviral therapy)

Dear Dr. Chien-Ching Hung:

Your manuscript has been accepted, and I am forwarding it to the ASM Journals Department for publication. You will be notified when your proofs are ready to be viewed.

Sincerely,

Yuan Pin Hung
Editor, Microbiology Spectrum
